# Combined Protein- and Ligand-Observed NMR Workflow to Screen Fragment Cocktails against Multiple Proteins: A Case Study Using Bromodomains

**DOI:** 10.3390/molecules25173949

**Published:** 2020-08-29

**Authors:** Jorden A. Johnson, Noelle M. Olson, Madison J. Tooker, Scott K. Bur, William C.K. Pomerantz

**Affiliations:** 1Department of Chemistry, University of Minnesota, Minneapolis, MN 55445, USA; joh09238@umn.edu (J.A.J.); olso6682@umn.edu (N.M.O.); 2Department of Chemistry, Gustavus Adolphus College, St. Peter, MN 56028, USA; mtooker@gustavus.edu (M.J.T.); sbur@gustavus.edu (S.K.B.)

**Keywords:** PrOF NMR, ^1^H CPMG NMR, BPTF inhibitor, *Pf*GCN5 inhibitor, 3D fragments

## Abstract

As fragment-based drug discovery has become mainstream, there has been an increase in various screening methodologies. Protein-observed ^19^F (PrOF) NMR and ^1^H CPMG NMR are two fragment screening assays that have complementary advantages. Here, we sought to combine these two NMR-based assays into a new screening workflow. This combination of protein- and ligand-observed experiments allows for a time- and resource-efficient multiplexed screen of mixtures of fragments and proteins. PrOF NMR is first used to screen mixtures against two proteins. Hit mixtures for each protein are identified then deconvoluted using ^1^H CPMG NMR. We demonstrate the benefit of this fragment screening method by conducting the first reported fragment screens against the bromodomains of BPTF and *Plasmodium falciparum* (*Pf*) GCN5 using 467 3D-enriched fragments. The hit rates were 6%, 5% and 4% for fragments binding BPTF, *Pf*GCN5, and fragments binding both proteins, respectively. Select hits were characterized, revealing a broad range of affinities from low µM to mM dissociation constants. Follow-up experiments supported a low-affinity second binding site on *Pf*GCN5. This approach can be used to bias fragment screens towards more selective hits at the onset of inhibitor development in a resource- and time-efficient manner.

## 1. Introduction

Fragment-based drug discovery (FBDD) is a validated technique to uncover inhibitors for many protein classes and has resulted in four FDA-approved drugs [1,2,3,4]. Compounds in FBDD libraries are typically of low complexity and low molecular weight (average MW ≤ 300 g/mol), allowing these libraries to more effectively sample chemical space, with smaller library sizes than traditional high throughput screening libraries (average MW ≤ 500 g/mol) [5]. Fragments with more complexity via sp^3^ carbons, saturated ring systems, and stereocenters (i.e., 3D fragments) are also being investigated for providing increased diversity of fragment libraries [6,7,8,9,10,11]. The 3D-enriched fragments offer a potential solution to improve selectivity and physicochemical properties at the beginning of the drug development process [7,12,13,14,15,16,17]. Fragments often have weak affinity for their target protein, making it essential to use screening assays that can detect compounds with affinity values in the high μM to low mM range.

Ligand- and protein-observed NMR binding assays are effective fragment screening techniques. A 2019 poll by *Practical Fragments* listed ligand- and protein-observed NMR as the second and fourth most popular FBDD techniques, respectively [18]. These two NMR formats have complementary advantages [19]. Ligand-observed NMR assays such as Carr–Purcell–Meiboom–Gill (CPMG), saturation transfer difference (STD), and water–ligand observed via gradient spectroscopy (waterLOGSY) allow for the screening of fragment mixtures without the need for deconvolution experiments due to distinct ligand resonances. In practice, only a single protein is screened in a ligand-observed NMR experiment. For small proteins studied here (<15 kDa), experiment times can take 5–30 min. In contrast, protein-observed NMR techniques such as ^1^H-^15^N-heteronuclear single quantum coherence (HSQC) and protein-observed ^19^F (PrOF) NMR use distinct protein resonances allowing screening of multiple proteins in a single experiment. Although faster than CPMG NMR for small proteins (2–8 min [20]), protein-observed NMR methods require more protein for screening and additional time for resource-heavy deconvolution steps when screening mixtures of fragments.

Here, we take advantage of the benefits of ligand- and protein-observed NMR techniques by combining ^1^H CPMG NMR and PrOF NMR into a single screening workflow that allows testing fragment mixtures for binding against multiple proteins with minimal deconvolution experiments, which saves time and labeled protein. There are many examples of fragment screens using ^1^H CPMG NMR [21], STD NMR [21], waterLOGSY NMR [21], PrOF NMR [11,22], dual-protein PrOF NMR [23], and multi-protein HSQC NMR techniques [20,24]. From these NMR methods, we chose ^1^H CPMG NMR and PrOF NMR for this platform because a previous study showed low discrepancy between the assays [25]. Additionally, PrOF NMR gives significantly less complicated spectra than HSQC NMR, and is 2–3-fold faster [25,26], expediting experiment time and data processing. More than 70 proteins have been studied via PrOF NMR using more than 15 different fluorinated amino acids [27]. PrOF NMR has been used to study proteins of various sizes from the Fyn SH3 domain (7 kDa) [28] to the α7 single-ring of the 20S proteasome core particle from *Thermoplasma acidophilum* (180 kDa) [29], illustrating its wide applicability to different proteins.

To validate this approach, we screened a library of 467 3D-enriched fragments in mixtures of five against the bromodomains of BPTF and *Plasmodium falciparum* GCN5 (*Pf*GCN5). *Pf*GCN5 is a homolog of human GCN5, which is in the same bromodomain family (IV) of BPTF allowing early stage assessment of selectivity. A previous study using this 3D-enriched fragment library to target the bromodomain-containing protein BRD4 showed that the inclusion of 3D-enriched fragments into screening decks may be useful in uncovering novel, selective chemical matter for bromodomain inhibitor development [11]. Here, we employ the same library to target two different bromodomains in order to gain a better understanding of the broad applicability of using 3D-enriched fragments for targeting bromodomains.

Bromodomains are epigenetic effector domains that recognize N-ε-acetylated-lysine (Kac) on the N-terminal tails of histone proteins. Widely considered highly druggable [30], bromodomains present a useful set of target proteins to validate this workflow. When dysregulated, bromodomain-containing proteins have been associated with diverse diseases. In particular, the bromodomain of BPTF has been implicated in several cancers including melanoma [31], breast cancer [32,33], and high grade gliomas [33]. While there are two commercially available chemical probes (TP-238, NVS-BPTF-1) and three literature-disclosed BPTF inhibitors [34,35,36,37,38], they all suffer from poor physiochemical properties or poor selectivity against the other 60 human bromodomains. Thus, there is a need for new chemical matter to develop selective inhibitors for BPTF.

Non-human bromodomains have been identified in parasites carrying infectious diseases, including malaria. Epigenetic proteins, including bromodomain-containing proteins, regulate gene expression and replication of the parasitic cells during the sexual and asexual stages [39,40]. Thus, small molecule-mediated disruption of parasitic bromodomain interactions with acetylated lysines on histones is a promising avenue for stopping the disease progression. *Plasmodium falciparum (Pf),* the most lethal protozoan malarial parasite, has up to eight predicted bromodomains, two of which have been characterized—*Pf*GCN5 and *Pf*BDP1 [41,42,43]. Prior research aimed at targeting these bromodomains has focused on repurposing human bromodomain inhibitors. Consequently, the four lead inhibitors from these studies have <10-fold selectivity for parasitic cells over human cells, limiting their potential as therapeutics [44,45,46]. Given the selectivity challenge for targeting the BPTF and *Pf*GCN5 bromodomains, we rationalized that a dual-protein screen and the use of 3D-fragments would increase the odds of finding selective fragment leads in a time-efficient manner. To our knowledge, we report here the first fragment screens against BPTF and *Pf*GCN5 and conclude our report discussing the time and resource advantages of using this new screening workflow and its applications towards additional bromodomains.

## 2. Results

### 2.1. Screening Workflow

Screen preparation entails expression of unlabeled- and fluorine-labeled proteins and pooling a fragment library into mixtures. Fluorine incorporation is accomplished by metabolic labeling with fluorinated amino acids or amino acid precursors [27,47]. Aromatic amino acids are enriched at bromodomain binding sites and, as such, their fluorinated counterparts are good reporting resonances. Here, we use 5-fluorotryptophan (5FW)-labeled BPTF and *Pf*GCN5 as both proteins have a tryptophan proximal to the native Kac binding site in a region termed the WPF shelf (Trp2824 and Trp1379, respectively, Figure 1). *Pf*GCN5 has an additional tryptophan, Trp1454, distal to the binding site that gives additional surface coverage for binding site analyses and general information on protein behavior during the assay. Unlabeled proteins were used for the ^1^H CPMG NMR assay.

Previously, the replacement of W with 5FW in BPTF has been shown to impart only small structural perturbations and no effect on stability [36,48]. Similarly, circular-dichroism thermal-melt experiments show little structural perturbations and no difference in the protein stability from 5FW incorporation relative to the unlabeled *Pf*GCN5 bromodomain (T_m_
*Pf*GCN5/5FW-*Pf*GCN5 = 57.1 °C, Appendix A). ^1^H-^15^N HSQC NMR chemical shift comparative analysis of ^1^H-^15^N-5FW *Pf*GCN5 and ^1^H-^15^N-*Pf*GCN5 gave an average Δδ of 0.037 ± 0.024 ppm (see note in Appendix A) and a similar dispersion of resonances supporting a well-folded protein. Additionally, a PrOF NMR titration with bromosporine, a known high-affinity pan-bromodomain inhibitor, showed slow exchange kinetics of the WPF shelf resonance, indicating little perturbation of the fluorinated protein binding function (Figure S21). We further support a lack of significant perturbation from fluorination via comparing the affinity of a new fragment by PrOF NMR and ^1^H-^15^N HSQC NMR with the non-^19^F-labeled protein yielded less than a 2-fold difference in affinity. This analysis is described further in Section 2.3. The observation of only a modest perturbation to structure and function from fluorination is similar to our prior analyses of fluorinated bromodomains [25,49].

A 467-member 3D-enriched fragment library from Life Chemicals was used in this screen. In a previous study, we used this library to demonstrate that the use of 3D fragments is beneficial for bromodomain screening using the first bromodomain of BRD4 (BRD4 D1) [11]. The fragments were pooled into mixtures of 4–5 fragments to increase screening efficiency while minimizing spectral overlap in the ^1^H NMR spectra for CPMG NMR. We employ this library, and the same screening criteria here, to further understand the implications of 3D fragments for bromodomain screening.

The screening workflow consists of two dependent steps: 1) a PrOF NMR screen of the 5FW-labeled proteins and a cocktail of 4–5 fragments in a single NMR tube; 2) a ^1^H CPMG NMR binding assay to deconvolute the hit mixtures for each protein (Figure 1). In the first step, a dual-protein PrOF NMR spectrum was obtained in which 5FW-*Pf*GCN5, 5FW-BPTF (~ 50 µM each) and mixtures of fragments (400 µM individual fragment) were present in the same NMR tube. Similar to a previous dual-protein PrOF NMR screen with BRD4 D1 and BPTF [48], the resonances for 5FW-*Pf*GCN5 and 5FW-BPTF are resolved, allowing for monitoring protein-specific resonance perturbations (Figure 1). Chemical shift perturbations indicating a ligand binding event were based on a prior cutoff of Δδ ≥ 0.030 ppm as the definition for a hit mixture. This cutoff was based on a statistical analysis of chemical shift perturbations from a prior BRD4 D1 fragment screen [23] and are described further in the experimental section and in Appendix A.

The second step is a ^1^H CPMG NMR deconvolution of the hit mixtures using the non-fluorine-labeled bromodomains. ^1^H CPMG NMR employs a T_2_ filter to take advantage of the difference in transverse relaxation time between protein and small molecules to filter out the bound-state ligand(s). Unlike the PrOF NMR assay, a single protein is assayed in each experiment. A binding event is characterized by a decrease in ligand resonance intensity from the ligand alone spectrum to the ligand + protein spectrum. A hit is defined as a decrease in resonance intensity ≥ 20%. The ^1^H CPMG NMR assay can be tuned by changing the protein or ligand concentration, filter length (Figure 1), or hit criteria depending on how stringent the desired hit rate is. The number of ^1^H CPMG NMR deconvolution experiments is directly informed from the first step. If a mixture from the first step was only a hit for one protein, a single ^1^H CPMG NMR deconvolution experiment is needed. If a mixture from the first step was a hit for both proteins, two separate ^1^H CPMG NMR deconvolutions experiments are needed to determine the fragments that bind to each protein.

### 2.2. Screening Results

In the first step, PrOF NMR gave a total of 49 out of 98 (50%) hit mixtures, with Δδ ranging from 0.030 to 0.46 ppm (Appendix A). There was an even distribution of hit mixtures binding to BPTF or *Pf*GCN5, and those that bind to both proteins (Table 1). Similarly, the ^1^H CPMG NMR second deconvolution step gave an even hit rate distribution of fragments that bind selectively to BPTF or *Pf*GCN5. An analysis of the range of PrOF NMR Δδ and ^1^H CPMG NMR percent resonance decreases showed no major differences between the proteins, indicating a similar responsiveness to binding (Appendix A). Additionally, no obvious trends were observed between the magnitude of the PrOF NMR Δδ and the magnitude of the ^1^H CPMG NMR response or the number of hits per mixture (Appendix A).

There was a 16% (8 mixtures) discrepancy between the steps of the assay where a hit mixture was indicated by PrOF NMR, but no hit fragment(s) was detected via the ^1^H CPMG NMR deconvolution experiment. A similar low disagreement (15%) between PrOF NMR and ^1^H CPMG NMR has been described by Urick et al. [25]. Six of eight mixtures leading to the discrepancies are likely due to the lenient cutoff used to define a PrOF NMR hit. These six mixtures had a Δδ < 0.050 ppm, indicating that these could be false positives. More surprising, the remaining two mixtures had a Δδ > 0.100 ppm but gave no ^1^H CPMG NMR hits. This result may indicate enhanced binding of fragments due to fluorination. An alternative hypothesis is that these mixtures contain high-affinity binders that show a limited signal response in the ^1^H CPMG NMR assay due to a lack of sufficient chemical exchange, a limitation of the ^1^H CPMG binding assay [50]. However, this effect was not pursued in this report.

Previously we used the same 3D fragment library in a screen targeting BRD4 D1 [11]. The 34 hits from that screen were also tested for selectivity against BPTF using PrOF NMR, giving 13 fragments that bound BPTF in addition to BRD4 D1. In cross-validation of our data, we identified 11 of these 13 fragment hits in this screen, an 85% agreement between the two screens for fragment hits binding to BPTF (Appendix A). The low discrepancy observed between the PrOF NMR and ^1^H CPMG NMR assays in this screen and the reproducibility of the PrOF NMR data when compared with our prior screen, indicate that the PrOF NMR and ^1^H CPMG NMR assays are ideal for using in sequential order for a screening workflow.

### 2.3. Hit Follow Up

Nine fragment hits were chosen to be characterized—six that selectively bound *Pf*GCN5 (**1**, **2**, **3**) or BPTF (**4**, **5**, **6**) and three that bound both proteins (**7**, **8**, 9**,**
Table 2). Fragments were selected to represent the broad range of PrOF NMR and ^1^H CPMG NMR magnitude of responses. Fragment **9** was included in the characterization because its hit mixture (#*30*) gave the highest and fourth largest Δδ (0.469 ppm, 0.169 ppm) for BPTF and *Pf*GCN5, respectively. Our previous screen identified **9** as binding BRD4 D1 with BPTF as an off target [11]. The single-point PrOF NMR assay with BPTF from the previous screen gave a Δδ of 0.453 ppm (400 µM **9**), similar to the Δδ observed with mixture #*30*, making **9** the most likely ligand from the fragment mixture. While a co-crystal structure of **9** with BPTF has been solved and a K_d_ for BPTF of 180 µM via PrOF NMR was determined [51], **9** has not been explored for targeting *Pf*GCN5.

PrOF NMR titrations of the selected fragments allowed for facile affinity determination. Hits that were selective for *Pf*GCN5 all showed a dose dependence; however, only 2 reached saturation, allowing for K_d_ determination. BPTF selective compounds gave a similar result with **6** but not **5** reaching saturation. While **4** selectively bound weakly to BPTF in the initial PrOF NMR screen and had strong binding response in the ^1^H CPMG NMR deconvolution assay, it showed no binding during the validation dose-response PrOF NMR experiment. Three other hits were found in mixture #*50* containing **4**, indicating that this false positive could result from ligand–ligand interactions affecting the screening assays or fluorine effects precluding binding. As expected, **7** and **8** bind BPTF and *Pf*GCN5, albeit with modest affinity. However, **6** is an attractive starting point for *Pf*GCN5 inhibitor development because it binds *Pf*GCN5 > 6-fold better than BPTF and was found previously to not bind BRD4 D1 (Table 2) [11]. Further, **9** is also an attractive target for *Pf*GCN5 inhibitor development because it has a good affinity and high ligand efficiency (K_d_ = 16 µM, LE = 0.43) and binds *Pf*GCN5 9-fold over BPTF. However, **9** also binds BRD4 D1 with a K_d_ of 50 µM [11].

### 2.4. Investigating a Possible Orthogonal Binding Mode to PfGCN5

The two tryptophans of *Pf*GCN5 gave additional biophysical information in the PrOF NMR assay that supported **9**, engaging a new low-affinity site on *Pf*GCN5. Typically for the PrOF NMR assays with bromodomains, the resonance displaying the largest response to a ligand binding is the 5FW near the native Kac binding site. Using a W1379F 5FW *Pf*GCN5 mutant, we assigned the downfield resonance in 5FW-*Pf*GCN5 as Trp1379 located on the ZA loop proximal to the Kac binding site in the WPF shelf (Figure 2, Appendix A). The downfield resonance consistently has the largest magnitude of response when tested with ligands known to bind in the native bromodomain binding site. Trp1454 is on the C helix located ~ 22 Å (C5-C5) from Trp1379. The response of both resonances may indicate a conformational change in the protein upon binding or ligands accessing a second binding site near Trp1454. Although rare for bromodomains, recently an additional binding site has been discovered for the second bromodomain of BRD4 [52]. Titration of **9** with *Pf*GCN5 showed significant responses of both 5FW resonances (Figure 2) and warranted additional exploration of a possible second binding site on *Pf*GCN5. Four other mixtures (#*32, #53, #51,* and *#94*) had a *Pf*GCN5 upfield Δδ > 0.030 ppm, indicating that other hits may behave like 9. To further explore the binding mode of **9** to *Pf*GCN5, several additional experiments were performed as described below.

The competition ^1^H CPMG NMR and PrOF NMR assays were employed to further probe the binding mode of **9** to *Pf*GCN5. L-Moses, a potent PCAF/*Pf*GCN5 inhibitor (K_d_
*Pf*GCN5 = 280 nM [46]), GSK4027, a BPTF/GCN5/PCAF inhibitor [53], and TP-238, a CECR2/BPTF chemical probe [54], were used as competitive molecules. Crystal structures of L-Moses with *Pf*GCN5 (PDB 5TPX), GSK4027 with GCN5 (PDB 5MLJ), and TP-238 with BPTF [51] show these compounds bind the native Kac site. An N-terminal tail peptide of a histone protein, H2A.Z II K7,13ac, was also employed as this peptide was previously shown to bind BPTF (K_d_ = 310 µM) [55] and also binds *Pf*GCN5 (K_d_ = 670 µM, Appendix A). Although the affinity for *Pf*GCN5 is unknown, a PrOF NMR titration confirmed that GSK4027 binds *Pf*GCN5 strongly and only perturbs the downfield resonance (Appendix A). In ^1^H CPMG NMR competition experiments, L-Moses and GSK4027 were unable to compete with **9** and keep it from binding to *Pf*GCN5 at low concentrations. At high concentrations, at or above the protein concentration, the competition maxed out at ~ 50% (Figure 2, Appendix A). Conversely, TP-238 was able to compete **9** off from BPTF at low concentrations. This result supports an additional binding interaction with *Pf*GCN5. In a different competition assay, PrOF NMR titrations that were used with a near stoichiometric concentration of GSK4027 while titrating in **9** showed several slow exchange resonances for both 5FW resonances (Figure 2). Interestingly, at high concentrations of **9** in the presence of GSK4027, the WPF shelf resonance, Trp1379, is at a new downfield position relative to the resonance without GSK4027 supporting an additional binding interaction that is stable in the presence of GSK4027 (Figure 2B).

To cross-validate our findings, ^1^H-^15^N HSQC NMR was used to further examine the binding mode of **9** to *Pf*GCN5 in the presence and absence of known ligands. The ^1^H-^15^N HSQC spectrum of *Pf*GCN5 is unassigned. For ease of spectral interpretation, all resonances that were distinctly visible were assigned as *1-64*. Fragment **9**, GSK4027, and the acetylated peptide H2A.Z II K7,13ac were titrated against *Pf*GCN5. At the highest ligand concentration, resonances with Δδ > *x^−^ +* 0.60**s*, where *x^−^* is the mean and *s* is the standard deviation at the highest concentration, were defined as significant. The correction factor of 0.60 is to compensate for only 60% of the resonances being usable for analysis. Titration of **9** with *Pf*GCN5 gave linear and non-linear dose-dependent chemical shift perturbations. Of the five resonances with non-linear movement, only one (*10*) was considered to have significant movement. Nine of the linear dose-dependent resonances were used to calculate a K_d_ average of 7.5 µM with a standard deviation of 7.0 µM (Appendix A). This affinity with the non-fluorine-labeled protein is similar but higher than the affinity determined by PrOF NMR. The resonances of *9, 30, 33, 55*, and *61* are common amongst **9**, GSK4027, and H2A.Z II K7,13ac, indicating that these are likely resonances affected by ligand binding the native Kac site. Both GSK4027 and **9** affect unique resonances (**9** = *4, 7, 18, 48, 53,* and *51*; GSK4027 = *5, 12, 20, 40, 43, 44, 50,* and *57*). It is likely that the unique resonances affected are due to its higher affinity and more protein-specific contacts because GSK4027 does not perturb the upfield resonance in the PrOF NMR assay. For fragment **9**, these resonances may be due to interactions outside the binding site. A competition experiment was performed where H2A.Z II K7,13ac was present at a concentration above the K_d_ (1 mM) and **9** was titrated. In this case, the resonances unique to **9** were affected as well as three additional resonances (*42, 49,* and *63*), supporting a possible ternary complex.

The competition ^1^H CPMG NMR, PrOF NMR, and ^1^H-^15^N HSQC NMR data support an additional low-affinity binding interaction of **9** to *Pf*GCN5. Finally, we used the in silico solvent docking program, FTMap, to identify any additional hot spot residues outside the native histone binding site [56]. This program has been used previously by Olp et al. to identify a new binding site on BRD4 D2 [52]. In this case, whereas several large solvent clusters were found in the histone binding site, only a single cluster of 12 solvent molecules was found near Trp1454. This result supports a potential binding site, but of low affinity (Appendix A). However, more biophysical data are needed to validate the hypothesis.

## 3. Discussion

### 3.1. 3D-Enriched Fragment Analysis

Due to drugs and chemical probes having a high degree of 3D character, 3D-enriched fragment libraries are increasingly being explored to generate starting chemical matter for inhibitor development [8,12,15,57,58,59,60]. The main concerns with using 3D-enriched fragments are the potential for lower hit rates due to increased complexity and hits that are biased away from fragments with 3D character limiting the utility of using these higher complexity molecules. Here, we sought to follow up on our previous study using 3D-enriched fragments to target BRD4 D1 [11]. In that screen, we observed a lower hit rate (10%) than a similar 2D-enriched screen (25%); however, the 3D character of the hits matched that of the overall library, indicating that both 2D and 3D fragments should be included in screening decks for increasing library diversity. In this screen, we observed a total hit rate for *Pf*GCN5 and BPTF of 9.8% and 9.2%, respectively, essentially the same hit rate as in the previous screen, indicating a similar ligandability between the three bromodomains, although BRD4 D1 ligands tended to be higher in affinity.

Plane of best fit (PBF) is a common method to quantify the 3D character of a compound [61,62]. While a PBF > 0.25 has been used by others to define a 3D fragment [62], we use a more rigorous cutoff of PBF > 0.30 as being considered highly 3D. The average PBF for all hits (0.42), BPTF selective hits (0.30), and *Pf*GCN5 selective hits (0.55) are above or equal to the 0.30 threshold (Figure 3A). Interestingly, *Pf*GCN5 selective hits are more 3D than BPTF selective hits. The non-parametric Mann–Whitney–Wilcoxon statistical test was used to analyze the PBF distributions. The overall hits and *Pf*GCN5 selective hits fall within the same PBF distribution of the library, indicating that the hits are not enriched for 2D or 3D character (Appendix A). Despite having an average PBF above the 3D character threshold, the distribution of BPTF selective hits is statistically different from the overall library, indicating that these hits are biased towards flatter fragments relative to the content of the overall library. In agreement with our previous conclusion, these results support including both 2D- and 3D-enriched fragments in screening decks for capturing more diverse screening hits which may improve the selectivity of fragment hits over structurally related bromodomains.

### 3.2. Initial Assessment of the Broader Applicability of the Screening Workflow

The dual-protein PrOF NMR, ^1^H CPMG NMR deconvolution screening workflow described herein can be broadly applicable to screening different targets. Although ^19^F is hypersensitive to its environment, making it a useful NMR active reporting nucleus, the main concern in screening two proteins simultaneously in PrOF NMR is the overlap of ^19^F resonances. We have screened pairs of proteins that are not members of the same bromodomain family (BRD4 D1 and BPTF [48]) and from different species (BPTF and *Pf*GCN5) in the same NMR tube with resolved resonances. Importantly for gaining selectivity information at the early stages in development, bromodomains that are members of the same family (IV), CECR2 and PCAF, have resolved 5FW resonances and can be used in a dual-protein assay (Appendix A). Moreover, a survey of nine 5FW-labeled bromodomains (BRD4 D1, BRD4 D2, BRDT D1, BRD2 D1, BRD2 D2, CECR2, PCAF, *Pf*GCN5, and BPTF) illustrates that 32 out of the 36 (89%) possible pair combinations of proteins have resolved resonances including different bromodomains on the same protein (Figure 3B, S9). If the desired target proteins have overlapping 5FW resonances, other ^19^F-labeled amino acids such as 3-fluorotyrosine, 4-fluorophenalanine, trifluoromethyl probes, or other isomers of fluorinated tryptophan (e.g., 6-fluorotryptophan) could be employed, as these resonances fall in a different region of the ^19^F NMR spectrum [63,64,65]. Additionally, if PrOF NMR cannot be used to screen multiple proteins at once due to overlapping resonances or perturbation of structure or function due to fluorinated amino acid incorporation, the RAMPED UP HSQC NMR [24] can be substituted for PrOF NMR in the protein-observed step of this workflow.

A concern with screening multiple proteins in the same solution is the possibility of non-specific protein–protein interactions that can influence ligand binding interactions. To explore this, we examined the chemical shift and line width at half height of the resonances for the three sets of proteins that have been tested in the same PrOF NMR experiment (BPTF/BRD4 D1, *Pf*GCN5/BPTF, CECR2/PCAF). The average change in chemical shift and change in line width of the resonances when the proteins were tested individual or in pairs was 0.04 ppm (0.0–0.08 ppm) and 4.3 Hz (0.4–9.2 Hz), respectively (Appendix A). If significant protein–protein interactions were occurring, we would expect a substantial change in chemical shift and line width of the resonances. Given the sensitivity of the ^19^F nucleus, low micromolar concentration of fluorinated protein can be used (25–50 μM), which may help to further mitigate non-specific interactions.

In comparison to similar screening methods, the dual-protein PrOF NMR screen and subsequent ^1^H CPMG NMR deconvolution is time and resource efficient while giving more biophysical information. There are several variations on the screening workflow described here to accomplish a fragment screen against two proteins—two separate ^1^H CPMG NMR screens, two separate PrOF NMR screens with PrOF NMR deconvolution experiments, or a dual-protein PrOF NMR screen with PrOF NMR deconvolution (Table 3). Conducting two separate ^1^H CPMG NMR screens is the most material efficient and the second most time efficient. However, all ligand-observed methods do not provide information on protein behavior during the assay or binding site location without additional competition experiments increasing time and material consumption. This added structural information is a general advantage of protein-observed NMR experiments. Because alternate PrOF NMR screening formats require multiple experiments for deconvolution of hit mixtures, this method is ~ 2.5-fold faster, uses 1.5–2-fold less ligand, and 1.3–1.8-fold less fluorine-labeled protein. In the case where a mixture only binds one protein in the PrOF NMR screen, the same NMR tube could be used for the ^1^H CPMG deconvolution, further reducing time and resource consumption.

In silico screening could be incorporated into this workflow in order to increase the amount of chemical matter explored. Virtual screening has been used to support and conduct fragment screens [66,67]. In particular, virtual fragment screens have been conducted against various bromodomains including BAZ2A [68], BET family bromodomains [69,70], and CREBBP [71]. In this workflow, large fragment libraries could be first screened against the two proteins of interests filtering the libraries into smaller libraries. The initial in silico hit libraries can then be subjected to this workflow.

## 4. Materials and Methods

### 4.1. Materials

The 3D-enriched fragment library was purchased from Life Chemicals and divided into mixtures of 4–5 fragments as previously described [11]. Amino acids, uracil, thiamine-HCl, LB broth, biotin, and nicotinic acid were purchased from RPI Corp. The 5-fluoroindole, magnesium sulfate, succinic acid, calcium chloride, Iron(III) chloride, ^15^N ammonium chloride, L-Moses, and imidazole were purchased from Millipore Sigma. Potassium diphosphate, potassium monophosphate, sodium phosphate, manganese (II) chloride, zinc(I) chloride, and sodium chloride were purchased from Fischer Scientific. Cobalt (II) chloride, Ethylenediaminetetraacetic acid, and copper (II) chloride was purchased from Acros Organics. Boric acid was purchased from Mallinckrodt. Deuterium oxide and dimethyl sulfoxide-d6 were purchased from Cambridge Isotope Laboratories. GSK4027 was purchased from Cayman Chemicals and TP238 was synthesized previously [55].

### 4.2. Protein Expression and Purification

5FW-labeled proteins. The 5FW-*Pf*GCN5-his6 and 5FW-BPTF-his6 were expressed as previously described [36]. The protein expression plasmid for expressing W1379F 5F2-*Pf*GCN5-his6 was purchased from Genesrcript.

Unlabeled *Pf*GCN5 and BPTF. Unlabeled *Pf*GCN5 and BPTF were expressed as follows. The pNIC28-BSA4 (kanomycin resistance) encoding BPTF and pET15-MHL (ampicillin resistance) encoding *Pf*GCN5 were gifts from Stefan Knapp (Nuffield Department of Medicine, University of Oxford) and Ray Hui (University of Toronto), respectively. *E.coli* strain BL21(DE3) was transformed with the plasmid of the desired protein sequence and plated onto an agar plates containing the respective antibiotics. The plated cells were incubated at 37 °C overnight. A single colony was selected from the agar plate and inoculated in 5 mL LB media and the appropriate antibiotic (100 mg /L). The primary culture was grown overnight at 25 °C with shaking at 250 RPM. The primary culture was transferred into 1 L of LB containing the appropriate antibiotic (100 mg/L). The secondary culture was allowed to grow at 37 °C with shaking at 250 RPM until the optical density at 600 nm reached 0.6–0.8. The temperature was reduced to 20 °C for 30 min before inducing protein expression with 1 mM isopropyl β-D-1-thiogalactopyranoside. Cells were harvested after 12–16 h.

^15^N and ^15^N, 5FW-labeled *Pf*GCN5. Following the procedure for unlabeled *Pf*GCN5, when the secondary reached an optical density at 600 nm of 0.6–0.8, cells were harvested via centrifugation and resuspended in minimal media (33.7 mM disodium phosphate, 22.0 mM potassium phosphate, 8.55 mM sodium chloride, and 1g/L ^15^N ammonium chloride) with trace elements (134 µM ethylenediaminetetraacetic acid, 31 µM iron(III) chloride, 6.2 µM zinc(III) chloride, 0.72 µM copper(II) chloride, 0.42 µM cobalt(II) chloride, 1.62 µM boric acid, and 0.081 µM manganese(II) chloride), magnesium chloride (1 mM), calcium chloride (0.3 mM), biotin (1 µg/L), thiamine (µg/L), and glucose (0.4% *w*/*v*). For ^15^N/5FW labeling, 100 mg/L of 5-fluoroindole was added. Cells were allowed to equilibrate at 37 °C, 250 RPM for 1 h. The temperature was reduced to 20 °C for 30 min before protein induction with 1 mM isopropyl β-D-1-thiogalactopyranosid. Cells were harvested after 12–16 h.

Protein Purification and Characterization. Protein purification was accomplished as previously described. Quadrupole time-of-flight (Q-TOF) LC/MS was used to confirm the identity of the protein and determine percent fluorine incorporation. The accuracy of the instrument is 0.001% or an error in mass of ±0.001% (±1.5 Da for a 15 kDa protein). Percent fluorine incorporation was calculated using Equation (1), where 1 F is the peak intensity for the mass corresponding to the incorporation of 1 fluorine, etc.
(1)$$\%\;fluorine\;incorporation=\;\frac{\left(0\;F\;protien\;\times \;0\right)\;+\;\ldots \left(n\;F\;protien*n\right)}{\left(0\;F\;protien*n\right)\;+\;\left(1\;F\;protien*n\right)\;+\;\ldots \left(nF\;protien*n\right)}\;\;\times \;100$$

Calculated and observed masses for 5FW-*Pf*GCN5his6, 5FW-BPTFhis6, unlabeled *Pf*GCN5 and unlabeled BPTF: 16922 and 16921 kDa (85–95% ^19^F incorporation); 14719 and 14720 kDa (85–95% ^19^F incorporation); 14740 and 14739 kDa; and 16904 and 16092 kDa, respectively For ^15^N, 5FW *Pf*GCN5 mass spectral analysis proved difficult for accurate percent labeling determination, but PrOF NMR shows moderate incorporation (Appendix A) and the lack of Trp peaks in the ^1^H-^15^N HSQC spectrum (Appendix A) indicate that it is sufficient for comparing the secondary structure of unlabeled of 5FW-labeled *Pf*GCN5.

### 4.3. NMR Methods

PrOF NMR. PrOF NMR spectra were obtained on a Bruker Avance III HD 500 with the 5 mm Prodigy TCI inverse cryoprobe (^19^F S:N 2000:1). All PrOF NMR experiments had 5% (*v*/*v*) D2O, 0.42% trifluoroacetic acid (TFA), and the desired fragment(s) dissolved in DMSO. The DMSO concentration for all experiments was kept constant at 1%. TFA (−76.55 ppm) was used to reference the spectra. For the PrOF NMR screen, 5FW-BPTF and 5FW-*Pf*GCN5 (35–40 µM in 50 mM phosphate, 100 mM NaCl, pH 7.4) were tested with mixtures of 4–5 fragments, where each individual fragment was 400 µM. Parameters values were: relaxation delay = 0.7 s, acquisition time = 0.05 s, sweep width = 10, and transmitter frequency offset = −125 (for binding experiment) and −75 (for TFA reference experiment). A 10 Hz line broadening was applied. Titrations were fit to Equation (2), where *p* is the protein concentration and L is the ligand concentration.
(2)$$\Delta \delta =\Delta \delta_{max}\;\times \;\;\frac{(K_{d}+P+L\;)\;-\;\sqrt{\left(\left({\left(K_{d}\;+\;P\;+\;L\;\right)}^{2}\right)\;-\;4PL\right)}}{2P}$$

A PrOF NMR binding event is characterized as a change in chemical shift (Δδ) from the protein-only spectrum to the protein + fragment mixture spectrum. We used a prior cutoff of Δδ ≥ 0.030 ppm as the definition of a hit mixture, based on a statistical analysis of chemical shift perturbations from a prior BRD4 D1 fragment screen [22]. This cutoff was determined to be reasonable given the average Δδ of the non-WPF shelf resonance of 0.017 ppm for *Pf*GCN5 observed in this screen. Although the binding site residues are more significantly perturbed, the average Δδ of the WPF shelf resonance was close to 0.030 ppm (BPTF = 0.049 ppm, *Pf*GCN5 = 0.043 ppm, Appendix A).

^1^H CPMG NMR. ^1^H CPMG NMR screen data were acquired on Bruker Avance III HD 500 with the 5 mm Prodigy TCI cryoprobe inverse cryoprobe (^1^H 2500:1 S/N). Competition ^1^H CPMG NMR data were acquired on Bruker 700 MHz Avance with a CryoProbe 5 mm TXI (^1^H S:N with H_2_O suppression (2mM sucrose) - 900:1) and excitation sculpting water suppression. Proteins were present at 10 µM in 50 mM phosphate, 100 mM NaCl, pH 7.4 in 100% D_2_O. In separate NMR tubes, a ^1^H CPMG NMR pulse was applied to a 1) mixture of ligands (individual ligands present at 100 µM for the screen and 50 µM for competition experiments) and 2) ligands + protein. For competition experiments, the ligands were used individually, and the competitor ligands were spiked into tube 2. ^1^H CPMG NMR parameters for the screen: NS = 128, acquisition time = 2 s, τ = 2.5 ms, and L = 120. For the competition experiments, τ was increased to 4.0 ms.

^1^H-^15^N SOFAST HSQC.^1^H-^15^N SOFAST HSQC spectra were obtained on a Bruker 600 MHz Avance NEO with a CryoProbe 5 mm TCI (^1^H, ^13^C, ^15^N, ^2^H) *w*/ *Z*-gradient using Bruker’s IBS_SOFAST.x method with 32 scans, 0.05 s acquisition time, and 0.2 s delay. Data were acquired at 3015.15 K and 50–52 µM protein in 50 mM phosphate, 100 mM NaCl, pH 7.4 and 5% (*v*/*v*) D_2_O. Data were processed with nmrDraw, nmrPipe, and NFRAM Sparky. Δδ^1^H-^15^N was calculated using Equation (3).
(3)$$\Delta \delta =\;\sqrt{{\left(\Delta \delta \;1H\right)}^{2}\;+\;{\left(\frac{\Delta \delta \;15N}{661}\right)}^{2}}$$

### 4.4. Peptide Synthesis

The H2A.Z II K7ac, K13ac peptide was synthesized using standard N-9- fluorenylmethoxycarbonyl (Fmoc) solid-phase synthesis methods on NovaSyn TGR resin (Novabiochem, 0.25 mmol/g) using a Liberty Blue automated microwave synthesizer and N,N- diisopropylcarbodiimide (DIC) and Oxyma for amino acid activation. The peptide was cleaved from the solid support in a mixture of 95/2.5/2.5 trifluoroacetic acid (TFA)/triisopropylsilane/water for 2–5 h followed by evaporation of solvent under a nitrogen stream. The crude peptide was precipitated into cold diethyl ether and purified by reverse-phase HPLC on a C-18 column using 0.1% TFA water and CH_3_CN as solvents (4–24% CH _3_CN gradient over 30 min). Peptide molecular weight was confirmed using an Ab-Sciex 5800 matrix assisted laser desorption ionization (MALDI) time-of-flight mass spectrometer. Peptide theoretical and observed masse and analytical HPLC trace can be found in Appendix A.

### 4.5. Docking

Open-source FTMap [72] was used to probe possible binding sites on *Pf*GCN5 (PDB ID = 4QNS).

### 4.6. Circular Dichroism

A Peltier-equipped temperature-controlled Jasco J-815 spectropolarimeter was used to probe the secondary structure of 5FW and unlabeled *Pf*GCN5 by scanning the far-UV range (200–260 nm) at 25 °C. For measurements, 10.4 µM unlabeled *Pf*GCN5 and 8.3 µM 5FW-*Pf*GCN5 (8.8 mM NaCl, 3.3 mM phosphate at pH 7.4) at a pathlength of 1 mm were used. Spectra were collected at a rate of 50 nm/min over–260 nm. Thermal melting temperatures were determined by monitoring the ellipticity at 222 nm though a temperature scan from 20 to 95 °C (60 °C/h).

### 4.7. FW PfGCN5 Resonance Assignment

The W1379F PfGCN5 mutant was purchased from GenScript. The W1379F 5FW-*Pf*GCN5 mutant was expressed using the method described above.

## 5. Conclusions

We presented a time- and resource-efficient screening workflow that uses PrOF NMR followed by ^1^H CPMG NMR deconvolution to screen mixtures of ligand against two proteins, completing two screens simultaneously. To demonstrate the applicability of this workflow, we completed the first reported fragment screens against the bromodomains of BPTF and *Pf*GCN5 using a library of 3D-enriched fragments. Moderate hit rates were observed for both proteins and hit follow up of a select set led to molecules with moderate selectivity in some cases and, in the case of **9**, a potential new low-affinity binding site on *Pf*GCN5. This workflow is broadly applicable to different protein targets and could serve to gain selectivity knowledge at the onset of inhibitor development.

## Figures and Tables

**Figure 1 molecules-25-03949-f001:**
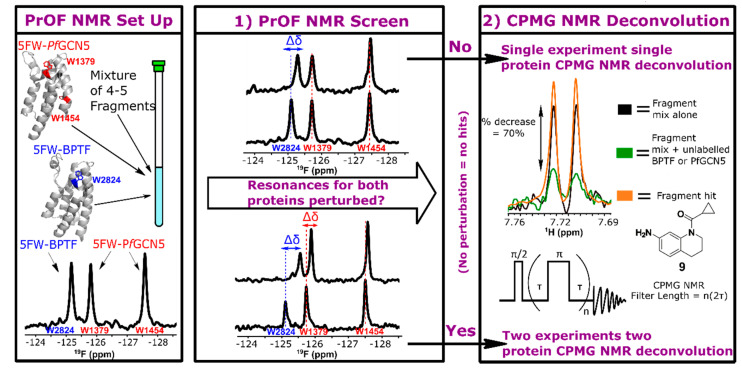
Screening workflow. (**Left**) The location of each 5FW resonance on the proteins and in the ^19^F NMR spectra (PDB ID = 4QNS, 3UVW). (**Middle**) Representative binding experiments in which case one or both proteins resonances are affected. (**Right**) A representative ^1^H CPMG NMR experiment in which case a decrease in resonance intensity is observed in the ligand + protein vs the ligand alone spectra. To deconvolute the ^1^H CPMG NMR mixture binding experiments, the ^1^H NMR spectra of each individual ligand in that mixture are overlaid to identify the corresponding ligand resonances.

**Figure 2 molecules-25-03949-f002:**
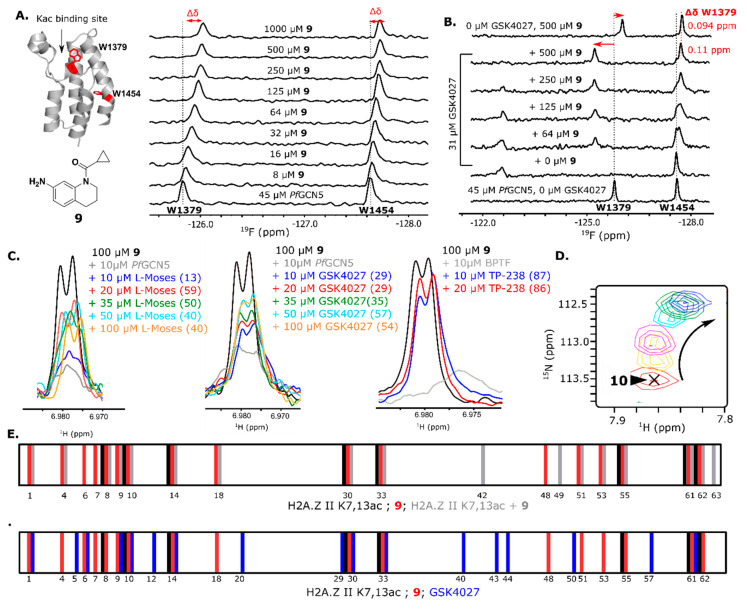
(**A**) Fragment **9** and *Pf*GCN5 (PDB 4QNS) with the W locations highlighted. PrOF NMR titration of **9** with *Pf*GCN5. (**B**) PrOF NMR titration of 9 with GSK4027 present at a saturating concentration. (**C)** Competition ^1^H CPMG NMR. The percent recovery for the addition of each competitor is shown in parentheses. (**D**) ^1^H-^15^N HSQC NMR non-linear movement of resonance *10* when titrated with **9**. The arrow indicates the direction of movement with increasing concentrations of **9** (0 µM = red, 16 µM = yellow, 32 µM = pink, 64 µM = teal, 125 µM = green, and 250 µM = blue). (**E**) Strip plots of *Pf*GCN5 ^1^H-^15^N HSQC NMR resonances with Δδ > *x^−^ +* 0.60**s* at the highest concentration tested (black = H2A.Z II K7,13ac, red = **9**, blue = GSK4027, and gray = **9** + H2A.Z II K7,13ac).

**Figure 3 molecules-25-03949-f003:**
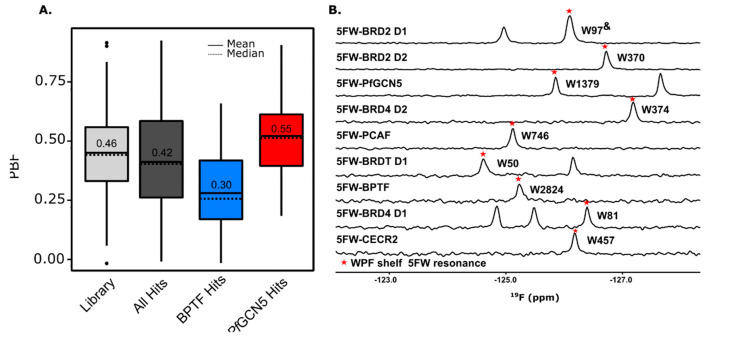
(**A**) Boxplot of the PBF of the library, total hits, and hits selective for BPTF and *Pf*GCN5. (**B**) PrOF NMR overlay of various 5FW-labeled bromodomains. The W in the WPF shelf near the binding site is denoted with a red star and label. ^&^ The 5FW resonances have not been assigned but this W is the one that shows the largest response with known ligands [55]. Also see Appendix A.

**Table 1 molecules-25-03949-t001:** Hit rate for both steps of the screening platform.

Protein(s)	Number of Hit Mixtures from PrOF NMR	Number of Hits after ^1^H CPMG NMR Deconvolution
BPTF	14 (14.2%)	27 (5.7%), (9.8%*)
*Pf*GCN5	16 (16.3%)	24 (5.1%), (9.2%*)
BPTF and *Pf*GCN5	19 (19.4%)	19 (4.1%)

* Total hit rate.

**Table 2 molecules-25-03949-t002:** Follow up on select fragment hits.

ID	Structure	Initial Mixture BPTF PrOF NMR Δδ	Initial Mixture *Pf*GCN5 PrOF NMR Δδ	^1^H CPMG NMR % Drop in Resonance Intensity BPTF	^1^H CPMG NMR % Drop in Resonance Intensity *Pf*GCN5	BPTF K_d_ (µM)	*Pf*GCN5 K_d_ (µM)	BRD4 D1 K_d_ (µM) ^b^
1	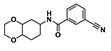	-	++	NA	+++++	NA	NS	NB
2	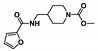	++	+++	NA	+++++	NA	360	450
3	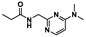 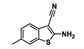	-	++++	NA	++++	NA	NS	NB
4	+	-	+++++	NA	NB	NA	NB
5	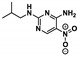	++	-	+++	NA	NS	NA	NB
6	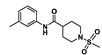 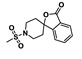	+	-	+	NA	540	NA	NB
7	+++	+++	++	+++++	> 1 mM	150	NB
8	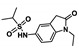	+++	++	++	++	720	> 1 mM	24
9	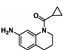	++++	++++	++++ ^a^	++++ ^a^	180	16 ^&^ (8.3, *s* = 7.0 *)	50

^a^ No initial ^1^H CPMG NMR performed during step two of the screen but ^1^H CPMG NMR was performed in competition experiments. ^b^ Data from previous screen [10]. For PrOF NMR = Δδ < 0.029 ppm, + = 0.030 < Δδ < 0.049, ++ = 0.050 < Δδ < 0.099, +++ = 0.1 < Δδ < 0.199, ++++ = Δδ > 0.2. For ^1^H CPMG NMR- = 0–19.9%, + = 20–39.9%, ++ = 40–59.9%, +++60–80%, ++++ = 80–99.9%, and +++++ = 100%. NS = non-saturating. NA = not applicable. ^&^ Determined as an average of W1454 and W1379. *Determined as an average of nine resonances via ^1^H-^15^N HSQC NMR titration (Appendix A).

**Table 3 molecules-25-03949-t003:** Comparison of the resources used in different screening methods based on the results of this screen.

	2 ^1^H CPMG NMR Screen	2 PrOF NMR Screens	Dual-Protein PrOF NMR Screen	This Method
**Screen Time (h) ** ^**a**^	30	21.7	10.9	10.8
**Deconvolution Time ** ^**a**^	0	36.7	36.7	10.8
**Amount of Protein Used for Screen (mg)** ^**b**^	8.5 (BPTF), 7.4(*Pf*GCN5)	42 (BPTF), 37 (*Pf*GCN5)	85 (BPTF), 74 (*Pf*GCN5)	85 (BPTF), 74 (*Pf*GCN5)
**Amount of Protein Used for Deconvolution (mg)** ^**b**^	0	74 (BPTF), 61 (*Pf*GCN5)	74 (BPTF),61 (*Pf*GCN5)	2.6 (BPTF), 2.1 (*Pf*GCN5)
**Amount of Fragment Used for Screen (mg)** ^**c**^	0.03	0.12	0.06	0.06
**Amount of Fragment Used for Deconvolution (mg)** ^**d**^	0	0.06 (bind 1 protein), 0.12 (bind both proteins)	0.06 (bind 1 protein), 0.12 (bind both proteins)	0.015 (bind 1 protein), 0.03 (bind both proteins)

^a^ Assumes a 9 min ^1^H CPMG NMR experiment and 6.5 min PrOF NMR experiment. ^b^ Assumes 10 µM per ^1^H CPMG NMR experiment and 50 µM each protein for PrOF NMR. ^c^ Assumes a molecular weight of 300 g/mol. ^d^ Assumes an average molecular mass of 300 g/mol, ^1^H CPMG NMR deconvolution of 100 μM per ligand, and PrOF NMR deconvolution of 400 μM per ligand.

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
