# Peer review of "Combined Protein- and Ligand-Observed NMR Workflow to Screen Fragment Cocktails against Multiple Proteins: A Case Study Using Bromodomains"

_molecules, 2020, doi:10.3390/molecules25173949_

Round 1

Reviewer 1 Report

In this work two fragment screening  NMR-based assays with complementary advantages have been combined into a new screening workflow. A benefit is for example that the method allows testing fragment mixtures for binding against multiple proteins with minimal deconvolution experiments. To validate this approach, authors screened a library of several fragments in mixtures of five against the bromodomains of BPTF and Plasmodium falciparum GCN5 (PfGCN5). The described approach seems promising and I have not particular issues against the manuscript. Some minor issues are reported below.

  1. The authors did not sufficiently explain how and why they selected fragments and targets to test the approach. In particular more emphasis should be given on why the above mentioned sample tests should be considered as a representative experiment for validating the procedure.
  2. Line 110 to line 135 seems to me a methodological part and not a Result.
  3. In the discussion section  some emphasis is given to 3D enriched fragments. The relationship between the novel workflow development and the use of this compound library should be better clarified, because the accent on this point creates a little of confusion.
  4. In silico docking is considered but not properly explained. Personally I do not think that an in silico method may be used to validate and experimental one… usually it is the contrary. I’m sure that few words more may better clarify the need for the in silico docking procedure.
  5. Advantages and limitation of the proposed workflow should be better emphasized. Maybe a table can be considered to show how the workflow of two techniques may overcome limitations of single separated techniques.
  6. In the manuscript attention is in general given to binding affinity and screening ability of the method. Can the authors provide more detail on any other relevant information that can be extracted?; e.g., structural information of binding sites, indications on moieties involved in the interaction site etc... (In this sense the in silico docking might be complemented to the experimental workflow. )
  7. A little discussion on how in silico methods may complement the novel workflow and/or what kind of information can be used by in silico simulation from the presented methodology would be very helpful for readers of this journal.

Reviewer 2 Report

Dear Authors,

The publication presented for review is very good. I am glad that you showed the possibilities of NMR technique for simultaneous screening of fragment cocktails against multiple proteins. I don't comment on your work.

Author Response

The publication presented for review is very good. I am glad that you showed the possibilities of NMR technique for simultaneous screening of fragment cocktails against multiple proteins. I don't comment on your work.

We thank the reviewer for evaluating this manuscript

Reviewer 3 Report

The present work suffers from methodological weakness. For this reason, it will be reconsidered after major revisions.

Major revisions:

  • I invite the authors to use a more specific and perhaps less ambitious title.
  • Please enforce the motivations that oriented you to use the present NMR techniques respect to others, despite the potential limitations in terms of protein production and possible experimental limitations (e.g. conformational perturbations).
  • If the authors claim to have succeeded in developing a useful workflow to apply to multiple proteins, it is necessary to validate it, introducing several controls, for example:
    • exclude the presence of protein-protein interactions
    • Intersect multiple experimental techniques that exclude a protein conformational modification after fluorine incorporation such that the binding ability of positive controls is not affected. PfGCN5 do not present any reference that justifies or supports this aspect. HSQC and CD are not sufficient in my opinion, in particular if the  main topic of the manuscript is the description of a workflow broadly applicable.
    • Please support by experimental data or motivate some crucial speculations (e.g. line 101, 156-157, 196-198, 200-202.
    • Some reported data did not support the idea that fluorine incorporation did not affect the protein function (line 156-157, 235-236 ecc..). Please provide a clarifications

Minor revisions:

  • Line please support with a proper reference

  • Figure 2b. Please check the last two spectra labelling (spectra named "+0uM 9" and "45uM PfGCN5")
  • Too many argumentations were reported in the results section. This aspect, in my opinion, weakens the idea of a robust workflow.

Round 2

Reviewer 3 Report

Majour revision

I again invite authors to use a more specific title, indicating the class of proteins under study. The modification provided is not sufficient, in my opinion.

As reported in their answers, they jutified every reviewer observations  supporting their answers on data obtained on a specific class of protein. It inderectly corroborates my doubt about generalization of the present method proposed. 

Please provide this modification.

Author Response

Reviewer 3:  

As reported in their answers, they justified every reviewer observations  supporting their answers on data obtained on a specific class of protein. It indirectly corroborates my doubt about generalization of the present method proposed. 

Although, we disagree with this point in part, as we felt we had indicated in the manuscript that the PrOF NMR has been applied to over 70 different proteins both in our lab and in others.   However,  given that the  experimental data  here was only applied to bromodomains here:  We have updated the title accordingly. 

Combined protein- and ligand-observed NMR workflow to screen fragment cocktails against multiple proteins: a  case study using bromodomains.